# Leadership and governance, financing, and coordination and their impact on the operationalization of health interventions in the humanitarian-development nexus in South Sudan

Amany Qaddour[1,2], Lauren Yan[3], Dan Wendo[4], Laura Elisama[4], Christopher Lindahl[5], Paul Spiegel[1]

**1** Department of International Health, Center for Humanitarian Health, Johns Hopkins Bloomberg School of Public Health, Baltimore, MD, **2** Rafik Hariri Center & Middle East Programs, Atlantic Council, Washington, DC, **3** Department of Mental Health, Johns Hopkins Bloomberg School of Public Health, Baltimore, MD, **4** MOMENTUM Integrated Health Resilience, Corus International/IMA World Health, Juba, South Sudan, **5** Consultant, Washington, DC

## Abstract

South Sudan ranks among the most fragile nations in the world. Protracted conflict and recurrent shocks have weakened its health system and contributed to high maternal and child mortality rates, despite large amounts of humanitarian and development assistance injected into the country since independence. Factors related to the leadership and governance, financing, and coordination of health services impact the feasibility of implementing the humanitarian-development nexus (HDN). Researchers employed a qualitative case study design drawing from document reviews and individual and group semi-structured interviews with humanitarian and development stakeholders in South Sudan (Juba capital and Bor town). Data was analyzed and findings were synthesized and organized into distinct themes. Forty-one interviews were conducted with 68 participants between November 2022 and January 2023, and 57 documents were analyzed. Findings showed that limited government investment in the health sector has perpetuated reliance on international assistance, and barriers to engagement with government counterparts have restricted coordination. Some nascent HDN coordination platforms exist with minimal political buy-in. Recent reductions in development health funding have complicated progress towards longer-term development objectives, including health systems strengthening. Structural barriers within multi-mandate agencies and differences in programming cycles, funding, and reporting contribute to silos. Continued fragility, a restricted operational environment, shrinking funds, and fragmented coordination have made it challenging to plan, finance, and implement HDN-health interventions. Informal efforts to bridge silos between humanitarian and development actors should become more formalized to use resources more efficiently. Despite certain restrictions in engagement with government, coordination and planning at the sub-national

**Data availability statement:** The qualitative datasets presented in this manuscript are not readily available given the confidential and sensitive nature of the data. These cannot be shared given interviews with key informants in South Sudan were conducted on the condition of full confidentiality and anonymity. Due to the sensitive nature of the interviews conducted as part of individuals' professional roles, anonymized summarized data may be available upon reasonable request to the Johns Hopkins Center for Humanitarian Health (email: humanithealth@jhu.edu).

**Funding:** This study is part of a broader project on the HDN and reproductive, maternal, newborn, child and adolescent health (RMNCAH) interventions in fragile settings by MOMENTUM Integrated Health Resilience under the MOMENTUM suite of awards, funded by the United States Agency for International Development (USAID). MOMENTUM Integrated Health Resilience is made possible by the generous support of the American people through a cooperative agreement with IMA World Health (Cooperative Agreement # 7200AA20CA00005). The contents of this article are the sole responsibility of the authors and the MOMENTUM partners and do not necessarily reflect the views of USAID or the United States Government. Website: https://usaidmomentum.org. The following authors received funding under this award (Cooperative Agreement # 7200AA20CA00005): AQ, LY, LE, DW, CL, and PS. USAID did not play any role in the study design, data collection and analysis, decision to publish, or preparation of the manuscript. LY received training grant support from the National Institute of Mental Health (NIMH), including NIMH grants T32MH103210, T32MH122357, and F31MH136678 during the research period. Website: https://www.nimh.nih.gov/. The NIMH did not play any role in the study design, data collection and analysis, decision to publish, or preparation of the manuscript.

**Competing interests:** The authors have declared that no competing interests exist.

level may still be feasible. Such engagement should be enhanced to ensure sustainability of health services. Investment in health systems strengthening and resilience by humanitarian and development actors should enable communities to absorb recurrent shocks and prevent backsliding in health provision.

## Introduction

After conflict spanning decades, South Sudan declared independence from Sudan in July 2011 [1]. Efforts have been underway to mediate intermittent conflicts, including the signing of various agreements and the establishment of the Transitional Government of National Unity in February 2020, which was extended to 2025 [1]. The United Nations Peacekeeping Mission for South Sudan was formally established in 2011, and still remains [2]. The Transitional Constitution outlined a decentralized structure consisting of the state, county, payam, and boma levels, however, authority and decision-making power are centralized at the federal level in Juba [3].

Since 2014, numerous targeted sanctions have been imposed by the United Nations (UN) and several governments, including the United States, on various individuals in government and the private sector involved in human rights violations, corruption, or obstruction to the Revitalized Agreement [4,5]. A lack of transparency by the government has consistently placed South Sudan in the top five countries of the Corruptions Perception Index since 2017 [6]. These factors have limited partnership with the government by international actors, particularly institutional donors. Donor restrictions for paying government personnel, often termed "zero-cash" policies, have had a critical impact on financial transactions with government line ministries [7].

In 2023, South Sudan was ranked third on the Fragile States Index [8]. Recurrent shocks and stresses continue to plague the population, including violence, poverty, flooding, droughts, acute food insecurity, and endemic and epidemic diseases [1,9]. The COVID-19 pandemic and looming threat of Ebola (since 2022) and Mpox have further exacerbated the fragility of its health system [9,10]. In 2023, nearly 2 million people were internally displaced in South Sudan and another 2.3 million people sought refuge outside of the country [11]. South Sudan also hosts over 330,000 refugees from neighboring countries, mostly from Sudan [11]. As a consequence of these compounded factors, development has been hindered and humanitarian needs are tremendous with over 6.8 million people (roughly 76% of the population) in need of such assistance as of 2023 (representing a 4% increase from 2022) [12,13]. Of these, 6.1 million people were in need of health assistance [12]. Health indicators are some of the poorest globally with maternal mortality at 1,223 deaths per 100,000 live births and under 5 years and neonatal mortality at 99 and 40 deaths per 1,000 live births, respectively (see S1 Table in S1 File) [14–18].

The Ministry of Health (MoH) formally governs the health system with national health policies in place [19]. Government contribution to the annual health budget averaged between 1–2% from 2011 to 2019 and commitments increased to 9.6% in 2021–2022, in part due to the COVID-19 pandemic, though there have been

concerns around whether the government will actually contribute what it has committed [20–22]. Current health budget allocations also represent a much lower commitment than what was outlined in the Abuja Declaration (2001), which entailed a minimum investment target of 15% by all countries in the African Union [23]. Under the MoH, the health system is decentralized across the state, county, payam, and boma levels of administration for health care levels including teaching hospitals, state hospitals, county hospitals, primary health care centers (PHCC), and primary health care units (PHCU) (Fig 1) [24]. The Relief and Rehabilitation Commission (RRC) oversees all humanitarian and disaster relief for the country, and functions as the local authority in the implementation of all interventions at the town, county, and state levels [25]. A summary of major health policies and plans is available in S2 Table in S1 File.

### Background on the humanitarian-development nexus and health in South Sudan

The complexity of fragility and protracted nature of the crisis in South Sudan continues to create challenges in the delivery of health services, including by humanitarian and development actors. The convergence of both humanitarian and development assistance within fragile settings—as is also the case in South Sudan—has been termed the humanitarian-development nexus (HDN, the nexus), which has been used to describe and guide the delivery of assistance in such settings, including their applications to the health sector [26–29]. Broadly speaking, the HDN can be defined as the overlap and intersection of humanitarian and development assistance and is intended to address the vulnerability of affected populations before, during, and after crisis [30]. The humanitarian-development-peace nexus (HDpN) or the "triple nexus"

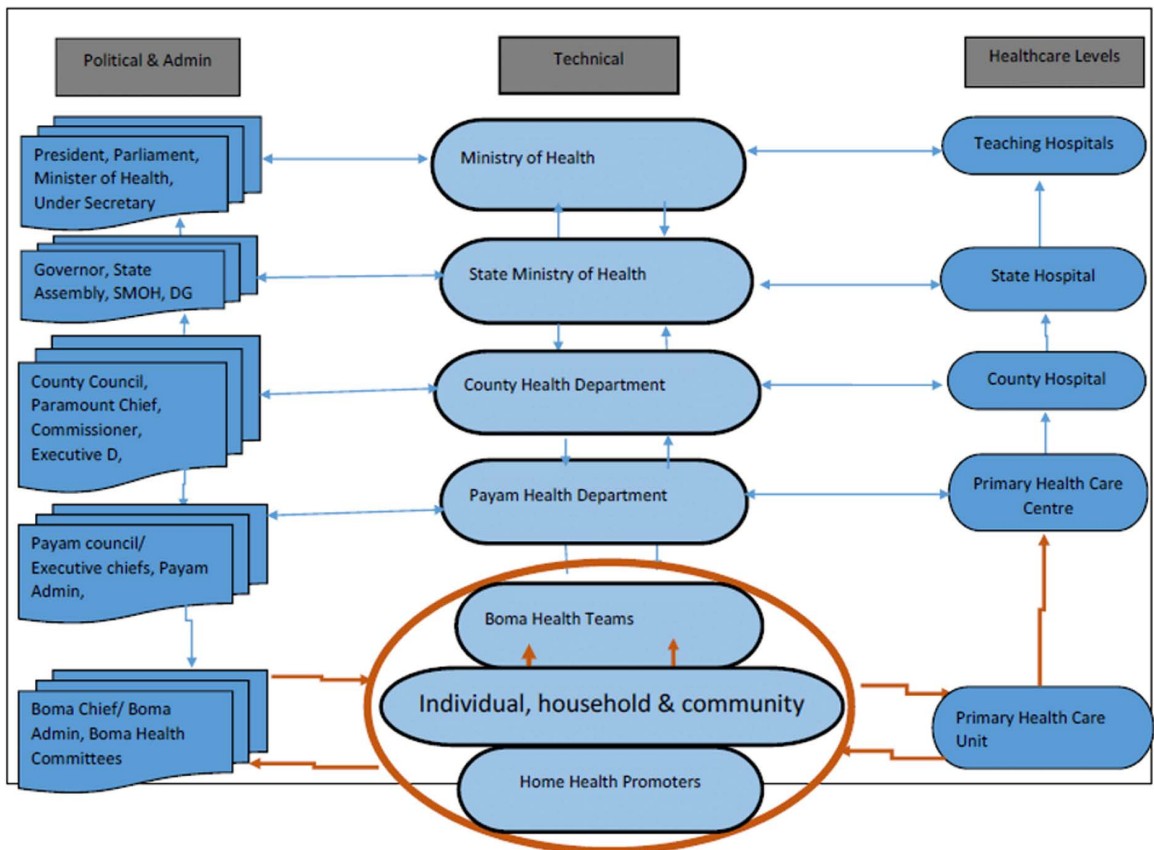

**Fig 1. Administrative health units of South Sudan.**

incorporates the concept of peace within the HDN. Measures to address drivers of fragility and root causes have contributed to the complexity of integrating the HDN and peace, with peace efforts often perceived to fall outside of the scope of humanitarian and development aid [31].

A number of distinctions exist between humanitarian and development assistance. Humanitarian assistance is primarily lifesaving and intended to "alleviate suffering and maintain human dignity during and after man-made crises and disasters caused by natural hazards, as well as to prevent and strengthen preparedness for when such situations occur" [32]. Four key principles govern humanitarian assistance, which are humanity, independence, impartiality, and neutrality [33]. Official Development Assistance is aid that "promotes and specifically targets the economic development and welfare of developing countries" [34]. The Sustainable Development Goals (SDGs) serve as the overarching guide for development, and consist of 17 goals, including for health, which falls under SDG 3 [35]. Table 1 summarizes the major distinguishing features of humanitarian and development assistance, which will serve as essential background information for this article (adapted from Andre Griekspoor) [36].

This research draws from a number of different frameworks and theories of change for the nexus and health. These include the health systems building blocks framework, established by the World Health Organization, which structures health systems into six different components, referred to as "building blocks," which include the following: 1) service delivery, 2) health workforce, 3) information, 4) medical products, vaccines, and technologies, 5) financing, and 6) leadership/governance [37]. Within the context of this research paper, the use of "health systems strengthening" terminology refers to one or more of these six health system building blocks. This research was also guided by the HDN-health conceptual framework, which we developed for the parent study (Fig 2) [38]. The HDN-health conceptual framework incorporates different elements from various guiding frameworks and organizes them into a set of core components: leadership & governance; coordination, financing, analysis & planning, and information management. These components build on the distinctive features of humanitarian and development assistance, which were outlined in Table 1 previously. In addition, a number of cross-cutting themes were embedded within the framework which intersect with health service delivery and contribute to the contextual nuances within a given fragile setting, including the application of humanitarian and human rights principles [39,40]; consideration of the various norms that may impact the implementation of health services, such as sociocultural, religious, and gender norms [41]; and localization commitments, to ensure local actors and affected communities are engaged in the design and delivery of health services, and actually leading these efforts, when feasible [42,43].

Health financing in South Sudan is complex, as humanitarian and development assistance has been injected into the health sector since 2011, and prior. This includes the parallel development health funds currently in place by the Health Pooled Fund (HPF) and the World Bank, in addition to humanitarian funding allocations. A timeline of these funds listed by source and designation is provided in S3 Fig in S1 File. The coordination mechanism consists of various fora, including

**Table 1. Distinguishing features between humanitarian and development assistance.**

|  | Humanitarian | Development |
|---|---|---|
| *Culture Approach* | Lifesaving and gap filling | Complementary to government |
| *Timeline (on average)* | 6-12 months, 2 years (multi-emergency awards) | 5-10 years |
| *Coordination* | System-led, clusters and sectors | Government-led; International Health Partnerships and related initiatives; Universal Health Coverage |
| *Planning Frameworks & Tools* | Humanitarian Response Plan (HRP); Refugee Response Plan (RRP) | United Nations Development Assistance Framework; Common Country Analysis; National Health Plan |
| *Legal Frameworks* | Humanitarian Principles; International Humanitarian Law | Sovereign Law; Aid Effectiveness Principles |
| *Types of Settings* | Fragile and insecure | Stable and willing |

   

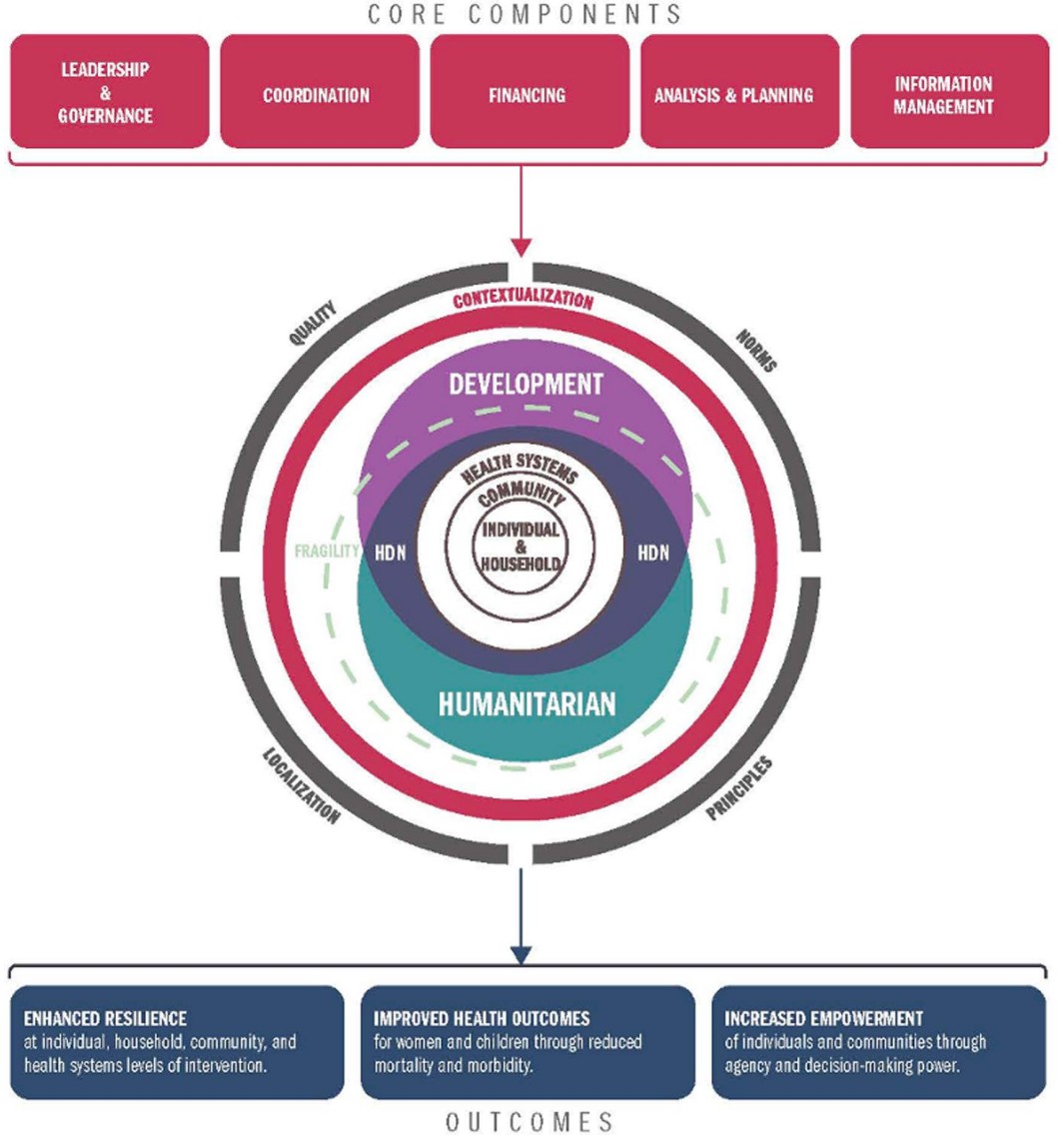

**Fig 2. HDN-health conceptual framework.**

the humanitarian cluster system by sector (e.g., health, nutrition), development coordination groups, and a select few hybrid platforms.

The aims of this paper are to: 1) analyze how leadership and governance, financing, and coordination influence and impact the operationalization of health interventions in the humanitarian-development nexus (HDN; also called the nexus) in South Sudan; 2) assess and document the barriers and challenges related to the three constructs and their intersection; and 3) analyze drivers and facilitators that enable the feasibility of implementing health interventions in the HDN. This study is part of a broader project on the HDN and reproductive, maternal, newborn, child and adolescent health (RMNCAH) interventions in fragile settings by the US Agency for International Development's MOMENTUM Integrated Health Resilience project (hereafter "MOMENTUM") [44]. While the aspect of peace is critical and relevant to the

implementation of the HDN in South Sudan, the scope of the research was on the HDN, rather than the HDpN. Therefore, the findings of this research will primarily focus on humanitarian and development aspects.

## Methods

### Study design and setting

We conducted a qualitative case study, drawing from document reviews and individual and group key informant interviews (KIIs). South Sudan was identified as a country case based on MOMENTUM's operational presence as a development actor working in the nexus on RMNCAH programming along with country office bandwidth. Research was conducted in Juba County of Central Equatoria state and Bor South County of Jonglei state [45]. Juba serves as both the capital and diplomatic center of South Sudan and Bor town is in Bor South County and is the capital of Jonglei State [46,47]. Bor has experienced violence before and after independence [48]. The area around Bor is primarily agricultural, producing grains and supporting livestock [46,49]. Jonglei state is prone to heavy rainfall and seasonal flooding each year and has experienced ongoing food insecurity since 2016 [50,51].

### Data collection

**Document review.** A document review was carried out prior to the country visit to sensitize the research team to existing policies, strategies, frameworks, assessments, and implementation plans for health/RMNCAH in South Sudan at the national and sub-national levels. Online sources for the document search included South Sudan's MoH portal, ReliefWeb, HumanitarianResponse.info, The World Bank, and UN agency sites. Publications which included a focus on RMNCAH and the HDN were included in the review. Documents that did not fall within the time-frame of 2011–2023 and did not include a RMNCAH-HDN focus were excluded from the review. Given HDN-specific terminology was not always explicitly used in documents retrieved, the review was guided by terms used in a previous HDN landscape analysis, including themes around fragility, sustainability, development plans, recovery, resilience, flexible funding, coherence between humanitarian and development assistance, transition, handover, localization, and quality [38]. All relevant content was extracted and organized in Microsoft Excel.

Documents were also obtained during the in-country visit from stakeholders. These included both public and non-public documents. All documents and resources obtained before and during the country visit were organized and used based on their relevance to the research question of this study. Criteria for inclusion of any documents or resources included the following: 1) information was relevant and within the scope of the research; 2) information provided useful background or context for the research team; and 3) and documents/resources helped the research team supplement, substantiate, or triangulate information provided during interviews. Peer-reviewed literature found through a concurrent scoping review on the HDN and SRMNCAH interventions in fragile settings (including South Sudan) were reviewed for this study. The scoping review protocol was registered in the Open Science Framework in October 2022 [Protocol DOI: https://doi.org/10.17605/OSF.IO/6BSE4] [29].

**Key informant interviews.** A KII guide was developed on RMNCAH and the HDN, drawing from the literature on interview guides and the HDN conceptual framework [37,38,52,53]. The process was iterative, as each section of the guide underwent extensive review during its development by the research team, a select group of external subject matter experts, and MOMENTUM's technical and country counterparts. Interviews were conducted on-site in Juba and Bor. All interviews were conducted in English by preference of interviewees, except one group KII conducted in the Dinka language, with on-site translation. After obtaining oral consent, interviews were audio-recorded and transcribed in real-time using the artificial intelligence transcription program Otter, followed by transcript review and validation support from Empire Caption Solutions and TranscriptionWing for accuracy [54–56].

A purposeful sampling approach was undertaken to select individuals and organizations at the capital and county levels [52]. Criteria were developed to ensure a diverse set of interviewees, which included individuals from humanitarian and

development organizations operating in South Sudan who were involved in health policymaking, planning, financing, and/or implementation. Target stakeholders who fit these criteria included government/MoH, UN agencies, institutional donors, local and international non-governmental organizations (L/INGOs), and private sector and facility-based service providers [52]. Stakeholders who met the criteria were selected then stratified by type and geographic region (Juba and Bor). Subsequently, a snowball sampling approach was utilized to identify additional interview participants and incorporate new perspectives [52]. The research team used this flexible, iterative approach to ensure a wide range of actors were interviewed to gain a comprehensive understanding of the operational landscape. The planned target was interviews with a total of 20–26 participants or until saturation was reached where there was no longer new information presented [52].

**Analysis.** Content analysis methods were used to analyze interview transcripts, which entailed coding transcripts for specific themes, concepts, and terms [52,57]. The software Dedoose was used for coding and analysis [58]. An analytical codebook was developed through a deductive process where codes were based on the study aims, the conceptual framework, and the interview guide [38,59]. The codebook was then refined through an inductive approach following the initial coding phase of select transcripts, particularly those that offered the most nuanced perspectives [53]. The coding process remained iterative to ensure researchers captured the subtleties across transcripts and allowed for a significant degree of flexibility, along with room for novel insights.

Document review and primary qualitative research findings were analyzed separately at distinct phases of the research. These findings were synthesized by organizing findings from both data sources into distinct thematic areas that were to be included in the research. Data were also used to substantiate, or at times contradict, information provided by key informants. Information from documents and key informants were triangulated to provide a more complete understanding of the situation and to ensure credibility, dependability, and areas of significance for research purposes [52].

**Ethics.** The Johns Hopkins Bloomberg School of Public Health determined this project "Not Human Subjects Research" as defined by DHHS regulations 45 CFR 46.102 and, therefore, does not require institutional review board oversight. All participants provided consent prior to interviews conducted.

## Results

A total of **41 KIIs** were conducted with **68 participants** between November 2022 and January 2023, including 28 individual and 13 group KIIs. A total of 38 KIIs were conducted in-country (KIIs 1–21; KIIs 25–41) and three via Zoom call (KIIs 22–25) due to scheduling conflicts [60]. Table 2 provides a summary of interviews conducted, organized by geographic location, KII (individual/group), stakeholder type, and gender (M/F). Stakeholders interviewed included MoH officials, institutional donors, UN agencies and cluster leads, L/INGOs, private sector, and health providers/committees. MoH officials in Juba were unable to meet during the data collection period despite multiple attempts by the research team. A total of **22 documents** were identified in the pre-visit document review and **35 resources** were obtained during the in-country visit. The latter included both public and private documents and online resources, including assessments, contingency plans, strategies, health dashboards, and medical consumption data. Pre-visit documents and relevant information are summarized in S4 Table in S1 File.

### Two overarching themes in the nexus

**Rigid institutional structures and the silos within multi-mandate teams.** According to some interview participants, a major barrier to operationalizing the nexus in South Sudan was the rigid, institutional structures, namely in large, bilateral and multilateral agencies, including the UN, donors, and INGOs. This was particularly the case for dual- and multi-mandate organizations, meaning those that had a combined humanitarian, development, and/or peace mandate. Of those interviewed from these agencies, some respondents cited working within silos, despite the existence of an overarching organizational nexus strategy at the headquarter level for their respective agencies. For example, in the case of dual-mandate agencies operating in South Sudan, humanitarian and development teams (and units, at large)

**Table 2. Summary of interviews conducted in South Sudan.**

| Key Informant Interviews—Juba | Type | Total KIIs | Participants | | |
|---|---|---|---|---|---|
| | | | Male | Female | Total |
| Donor Agency | Individual | 4 | 4 | 0 | 4 |
| Donor Agency | Group | 3 | 3 | 7 | 10 |
| UN Agency and/or Cluster | Individual | 3 | 3 | 0 | 3 |
| UN Agency and/or Cluster | Group | 2 | 2 | 2 | 4 |
| International NGO | Individual | 3 | 3 | 0 | 3 |
| International NGO | Group | 2 | 7 | 4 | 11 |
| National NGO | Individual | 2 | 2 | 0 | 2 |
| National NGO | Group | 1 | 3 | 0 | 3 |
| Private Sector | Individual | 2 | 1 | 1 | 2 |
| Private Sector | Group | 2 | 3 | 1 | 4 |
| | *Sub-totals* | *24* | *31* | *15* | *46* |
| **Key Informant Interviews—Bor** | | | | | |
| Government/Ministry | Individual | 3 | 3 | 0 | 3 |
| Health Provider | Individual | 3 | 2 | 1 | 3 |
| Health Provider | Group | 1 | 2 | 1 | 3 |
| International NGO | Individual | 4 | 4 | 0 | 4 |
| International NGO | Group | 2 | 5 | 0 | 5 |
| National NGO | Individual | 3 | 3 | 0 | 3 |
| Health Committee | Individual | 1 | 1 | 0 | 1 |
| | *Sub-totals* | *17* | *20* | *2* | *22* |
| | | | | | |
| | *Total KIIs* | *41* | *Total Participants* | | *68* |
| | *Individual* | *28* | *Male* | | 51 |
| | *Group* | *13* | *Female* | | 17 |

were separate and did not always have formal measures in place to coordinate interventions among each other from a planning, implementation, or financing standpoint. Rather, informal collaboration was said to be occurring among individuals from humanitarian and development arms, but not necessarily at the broader institutional level. High-level engagement was cited as infeasible by some, given the major differences between mandates, notably in terms of programming life cycles, reporting requirements, and separate and earmarked sources of funding. These distinctions were perceived to limit more systematic nexus efforts. In addition to structural barriers, interviewees highlighted challenges in terms of a power dynamic between international and local actors, along with the commitment to localization—the empowerment of local actors and communities from affected populations to "have the resources and agency to address the challenges that impact them" [42]. Commitments to break structural silos and enhance commitments to localization were deemed unsustainable and personality-driven given the high turnover of international staff in South Sudan, often resulting in transient leadership initiatives.

**A blurred line between humanitarian and development mandates.** According to interviewees, the protracted nature of fragility in South Sudan has blurred the lines between humanitarian and development assistance. Specifically for health interventions, similarities were highlighted in the types of programs implemented by humanitarian and development actors, with the only distinction referenced in multiple interviews being the source of funding and period of coverage. Given the substantial humanitarian needs cited across the population, both types of assistance entailed the provision of basic and lifesaving health services. Respondents attributed the inability to cover basic services to the significant

development funding gaps that were occurring between 2022–2023 (see financing section). This further distorted the mandates of humanitarian and development actors given the continued fragility in the country. One development actor illustrates the expansion of their primary health care mandate to include humanitarian response due to emergencies like COVID-19 and Ebola:

> *"They [donors] know what our capacity is and what we can or what we cannot do, because we're contracted, at the end of the day, to deliver primary health within this country. We are not here as emergency [humanitarian] responders. We do in the event there is an emergency and there is in our area, but for argument's sake, Ebola, COVAX, etcetera, that landed on our plate..."*—Development Actor (KI13)

Interviewees cited the difficulty in implementing sustainable development health programs due to the absence of peace in the country, by drawing from the notion that there simply could not be "*sustainable development without peace and no peace without sustainable development*" as per the Sustainable Development Goals [35]. One interviewee highlighted the impact of health interventions on development through investment in "peace dividends," in other words, peace-promoting activities, including the equitable provision of services across communities which may facilitate harmony and social cohesion in the interim, given long-term, political solutions have not yet materialized in South Sudan.

**Three fundamental components of the nexus: leadership & governance, financing, and coordination**

**Leadership & governance.**

**Leadership & governance in the health sector—who's in charge here?** Engagement with the South Sudanese government by humanitarian and development actors is complicated given the indirect consequences of targeted sanctions on the health sector, particularly with the MoH. Nearly all interviewees identified this as a significant barrier that impacted humanitarian and development actors differently due to their respective mandates. Among humanitarian actors interviewed, there was a consensus that working directly with the government amidst sanctions could compromise their commitment to the humanitarian principles, particularly independence, impartiality, and neutrality [39]. Although many development actors interviewed confirmed their commitment to these principles, their engagement was less restricted given expectations to engage with the government for development activities. Despite the desire of most actors, especially humanitarians, to uphold these principles, this lack of direct engagement was criticized as an attempt to bypass government, as one development actor illustrated:

> *"We are also working in a context in which there is a history of exclusion. […] So, you have a situation in which IPs (implementing partners) are working in communities and the government don't even have a clue what's going on […] that pointed to a level of resentment and discontent on the part of state government that IPs come in, bypass them and start talking to communities."*—Development Actor (KI6)

The government was considered to have limited ownership of the health sector, as humanitarian and development health assistance was predominantly managed by international actors. This created a parallel leadership structure, with the MoH's role considered more symbolic by some interviewed due to its pronounced reliance on international assistance. Consequently, there appeared to be greater autonomy and decision-making by international actors relative to their MoH counterparts, which raised concerns around the power dynamic among stakeholders. One respondent attributed this differential to the limited capacity of the government, further exacerbating dependence on external actors and limiting the sustainability of health services:

> *"The missing part is government capacity because whatever you do, the endgame [is] to exit. That's the endgame, so the community, government will take over. Without active government, it is [impossible]*— Dual-Mandate NGO (KI7)

Sustainability concerns were heightened in light of the immense development funding changes in 2022–2023, with over 220 of 797 health facilities covered by the HPF closed across eight states, the majority of which provided RMNCAH services [61]. These closures were dramatic and sudden, and most interviewees reported fears around the potential impact on health outcomes in the short- and long-term. Despite agreement by the government to take over these facilities and transition plans in place for this to occur, the donors interviewed confirmed that this did not occur, as the government did not cover these facilities following handover. Since the research was conducted, a consolidated multi-donor trust fund was announced to replace the two separate health development projects, as will be outlined later in this paper. A number of barriers to this transition were identified, including limited annual investment by the government in the country's health budget with donors continuing to subsidize a significant portion of health services. Other factors mentioned included a lack of political will by the government to uphold takeover commitments, and more broadly, global funding shortages due to concurrent crises such as those in Ukraine and Afghanistan.

**A decentralized health system on paper—a top-heavy one in practice**   National health policies reviewed in this study outline a decentralized health system under the general oversight of the MoH, with decision-making authority and prioritization of health services occurring at the sub-national level (state, county payam, boma) [19,62–66]. While this was the case in theory, multiple respondents voiced frustration that most decision-making occurred at the capital (Juba) or headquarter (international) levels by government or international organizations, and did not always align with health needs identified at state or county levels. This discrepancy was attributed to a variety of factors, including concerns around the politicization of aid, nepotism, and cronyism by government officials. Other respondents pointed to an internationally led, top-heavy response as the reason for the inability for decentralization to successfully work, with donors, UN agencies, and INGOs at the helm, essentially perpetuating this top-down approach. The concentration of decision-making power and oversight by so many actors at the top of the hierarchy relative to the field level where services were being delivered raised concerns around the efficiency and sustainability of health interventions. Interviewees at the state level conveyed concerns with relying on subjective decision-making in determining which facilities to support, such as using personal relationships to drive these decisions rather than relying on objective criteria (e.g., population density levels, needs assessments, etc.). Respondents cited that such subjective criteria perpetuated inequities in the health system across different states and continued to undermine sub-national leadership. Both international humanitarian and development actors interviewed discussed the challenges they faced in the absence of a fully functional MoH and health system, with one respondent stressing the impact of such limitations on sustainable development:

> *"That's our responsibility being superimposed within a weak system and within a scenario where we are simply providing basic services […] So the healthcare system is not functioning as a system that can sustain development in a meaningful way."*—UN Agency (KI14)

In these same discussions, the localization agenda was raised as a vital element for sustainability. With the bulk of service delivery led by international stakeholders, actors at the local level, including LNGOs and community groups, perceived themselves to be situated at the bottom of the hierarchies in place. LNGOs interviewed described this dynamic and blamed the endless series of "capacity building" initiatives imposed on them by international partners, with local actors not considered capacitated enough to lead at a certain point, even at the local level. One respondent captures this frustration:

> *"Generally, in South Sudan, there's only one word they are talking about and that's that national NGOs doesn't have 'capacity.' This has been for more than 25 years […] Okay, but even international NGOs have been here more than 30 years, so they're even unable to build a capacity for the national. Because one of their roles is to build the capacity of the national. You look at this period and up to now, you're saying this person, this organization has no capacity. Now whose mistake is here, right?"*— National NGO (KI18)

In summary, various factors in South Sudan have led to international actors taking the lead in the health sector and limited government leadership and decision-making power. The lack of government involvement and capacity along with limited political will and restrictedengagement (e.g., sanctions) have created gaps in the sustainability of health services. These have also posed limitations in the feasibility of transitions from international actors to the government, as evidenced by facility closures following HPF handover in 2022–2023. Additionally, the sustainability of health programming in South Sudan has been raised in close connection with localization efforts to strengthen the capacity of local actors in order to take over the delivery of health services, with hierarchies between international and local stakeholders hindering these efforts.

### Financing

**Major development funding drawdowns—perpetuating humanitarian needs.** The period of research coincided with significant changes in health development funding, including the drawdown of HPF funding by donors and the inability of the government to fulfill their commitments to takeover, resulting in the closure of 220 health facilities (as mentioned above), and the transfer of coverage of one additional state from the HPF to the World Bank. The split of development programs among states between the HPF and World Bank has created a discrepancy in health service provision. Differences were suggested in terms of the comprehensiveness of health services offered and the quality of services provided. While the reasons for this discrepancy were less clear, there was speculation among some respondents that more secure financial resources from the World Bank enabled more comprehensive services and enhanced quality relative to the HPF. In addition, the International Committee of the Red Cross (historically considered a humanitarian actor) is one of the largest implementing partners of the World Bank in South Sudan and operates outside of the traditional humanitarian coordination architecture given its official observer status granted by the UN [67]. Despite this, the operational environment for the three World Bank-coverage areas in 2023 (Unity, Jonglei, and Upper Nile states) were also more volatile, insecure, and conflict-affected relative to those under the HPF [68]. Despite both of these funds being classified as health development funds, the majority of services covered under each were almost identical to those funded by humanitarian actors, and in turn, resembled humanitarian funding but over a longer-term period given the basic, lifesaving nature of services and fragility of the health system.

The combined impact of shrinking development funds and increased humanitarian needs between 2022 and 2023 prompted more humanitarian actors to fill substantial gaps in the health system. However, humanitarian funding also decreased from 2022 to 2023 despite the increase in needs, with only 53.6% of funding secured for the Humanitarian Response Plan in 2023 relative to 73.7% secured for 2022 [12,13]. An example of gap coverage for the loss of development funding was the South Sudan Humanitarian Fund managed by the UN Office of Coordination for Humanitarian Affairs, which is intended for short-term coverage of acute, lifesaving humanitarian needs based on the Humanitarian Response Plan. In 2022, this Humanitarian Fund was brought into the fold to support health interventions previously covered by development health donors. This turn of events has somewhat forced humanitarian and development donors to coordinate efforts to determine how best to use available resources efficiently (as will be discussed in the coordination section). One donor illustrates the significant impact of funding drawdowns on transition planning between humanitarian and development assistance and on the continuity of health services:

> "We were also trying to understand sort of transitions and phase out of certain [donors] so we've seen multiple conversations we've heard people saying, 'Oh, funding's cut. That's it, we're done' […] So just in terms of thinking how some of those services continue when a lot of service actors are phasing out, and how donors are also thinking about this? […] If it's at a facility level and also large scale. Honestly, it's a nightmare."—Donor (KI21)

**Efforts to integrate flexible and cross-cutting humanitarian and development funding—more predictability and clarity needed.** Both humanitarian and development assistance in South Sudan integrated some form of flexible funding

for rapid-onset emergencies, particularly recurrent flooding, food insecurity, and mass displacements. Such flexible funding allowed for the adaptation of programming and scale-up in order to meet the needs among affected communities to these common shocks and stresses. However, it was difficult for development actors interviewed to respond to the numerous and diverse emergencies, as the instability and unpredictability of these emergencies often required them to halt their operations given these events extended beyond their development mandate. In these instances, some confirmed that they simply withdrew as it was expected that humanitarian actors would take over service delivery in these regions given their mandate and ability to rapidly mobilize resources, as one development NGO notes:

> "We could say because when people were displaced, the other actors, the humanitarian actors, they were able to access those places […] Like I said, because of our [development] nature, we were not able to adjust and apply for humanitarian funding for us to access those places."—Development NGO (KI5)

The ability of development actors to provide humanitarian assistance according to changes in the context was described as a challenging and rocky process that was complicated, as the scale of integrated funding for emergencies has differed depending on the source of funding. For example, in the case of the World Bank and its existing health systems strengthening programs, two different forms of emergency funding can be triggered including the Contingent Emergency Response Component and the Crisis Response Window. The former was used to disburse USD 40 million in response to the COVID-19 pandemic, flooding, and Ebola in target locations, including Upper Nile and Jonglei States and the Pibor and Ruweng Administrative Areas. In comparison, it was observed that the HPF Emergency Preparedness & Response Fund annual envelope amounted to less than 1% (roughly 0.33%) of its entire funding envelope for the year (USD 200,000 from a total of USD 60 million) for all seven states of coverage. Other contingency funds were mobilized at the onset of emergencies according to humanitarian actors interviewed, including the Rapid Response Fund and Emergency Rapid Response Mechanism, which essentially allowed these actors to respond to a crisis within a crisis [69,70]. Some, though not all, respondents noted these humanitarian-sourced funds were triggered more expediently as they were expressly earmarked for this purpose. However, there appeared to be less clarity on how well coordinated these funds were in terms of ensuring efficiency and complementarity with existing health programs, while preventing duplication and overlap amongst humanitarian actors themselves, as well as between humanitarian and development actors.

### Coordination

**A fragmented coordination landscape—insufficient coordination between development and humanitarian actors.** A large number of diverse humanitarian and development health organizations operate in South Sudan. In the health sector, at least 111 health agencies (besides the MoH) were identified according to mapping efforts for 2023, including 6 UN agencies, 11 donors, 57 national NGOs, 31 INGOs, and 2 agencies with observer status (i.e., ICRC and MSF) [71]. While numerous coordination platforms existed among the humanitarian actors and fewer, but equally important coordination mechanisms were in place for the development players, there were limited platforms for humanitarian and development actors to coordinate together. Table 3 provides a list of these major coordination bodies. It is important to note that the RH coordination forum was the only topic specific coordination mechanism examined in this paper, as the parent study focused on RMNCAH. For this reason, other fora (outside of the Health Cluster) were not explored, such as HIV/TB, disease surveillance, etc.

Most respondents referenced some fragmentation between development and humanitarian actors, with one major donor citing this as the "*name of the game here*" in South Sudan. Though there were distinct and separate coordination fora for humanitarian and development actors, these were often attended by the same individuals or organizations given many identified themselves as multi-mandate. The humanitarian coordination mechanism has formally been in place primarily through national and sub-national clusters given the explicit coordination mandate for humanitarian response. Much

**Table 3. Key coordination fora in South Sudan.**

| Coordination Body | Type |
|---|---|
| Strategic Advisory Group | Development |
| Donor Working Group | Development |
| Donor Working Group | Humanitarian |
| Resilience Advisory Group | Humanitarian-Development |
| Reproductive Health Coordination Forum | Development |
| National Steering Committee | Development |
| Heads of Coordination/Heads of Mission (HoCs/HoMs) | Humanitarian |
| National & Sub-National Health Clusters | Humanitarian |

of this coordination has been guided by the annual Humanitarian Response Plan, which includes concrete programmatic and financial targets by geographic area. Despite this planning and forecasting of needs, there was speculation among donors and implementing agencies interviewed that there was still some duplication in both humanitarian and development assistance given the concentration of actors often operating in the same geographic areas. In terms of coordination for development actors, this was less formalized with meetings occurring on a more ad hoc basis.

Although the UN Sustainable Development Cooperation Framework for South Sudan (2023–2025) outlines development targets and financial resources needed, this document was not referenced as a guide by any development actors interviewed in the same way the Humanitarian Response Plan was used as the overarching framework for humanitarians. Furthermore, the coordination for development was not donor-mandated, unlike humanitarian coordination, according to those interviewed from the development sector, but rather encouraged, particularly at the community level. However, the intense competitive environment of securing contracts by development actors was cited as a major barrier to effective coordination. Given the recent development health cuts, there appeared to be more explicit efforts to coordinate between development donors.

Due to changes in development funding and the enhanced role of humanitarian donors to address these gaps, broader nexus coordination efforts were also organized in 2022 by bilateral donors, such as Germany, in order to bring together humanitarian and development agencies. Although less formal, it appeared these meetings were to be planned on a more regular basis. A more high-level HDN coordination body was also in place, which brought together major international actors, though this body did not entail any joint mapping efforts by humanitarian and development actors in terms of charting activities implemented in each area of operation. Additionally, the UN Deputy Secretary General/Resident Coordinator/Humanitarian Coordinator had taken strides to bring together humanitarian, development, and peace actors building on the existing Partnership for Peace, Resilience, and Recovery initiative, which was revamped in 2022 [72]. Although this relaunch was still in its infancy at the time of our interviews, both humanitarian and development actors who referenced this initiative identified it as a potential facilitator of enhanced cooperation between actors across the nexus. Among respondents, however, there was not a clear picture on the extent of funding and political commitment for this effort.

**Health coordination and gaps in data sharing, decision-making, and prioritization of assistance.** Respondents cited flooding, COVID-19, and the potential Ebola outbreak, among others, as key inflection points in the coordination of health services in South Sudan, challenging both humanitarian and development actors' ability to coordinate more effectively, specifically in terms of preparedness, planning, and implementation. Although numerous coordination health meetings took place at the national level, these were cited as information-sharing in nature, rather than action-oriented or rooted in decision-making. Coordination at the state and county levels was focused on health services (primarily technical in nature), but there were concerns relayed around how much these platforms were taken into account when decisions were made at the national level. There was also less clarity in how services were coordinated and prioritized in areas that were deemed more stable relative to highly fragile regions that required more resources and immediate attention. A small

number of participants shared examples of how such fragility considerations informed resilience programming, especially in areas primed for early recovery and longer-term development assistance.

There were also major challenges in coordination at the state and county levels reported as it pertained to data availability and sharing, with interviewees referencing delayed or incomplete reporting due to the limited capacity, availability, and motivation of MoH staff. Reasons cited for this included MoH staff not being paid sufficiently or regularly relative to health cadre employed by international actors. Other challenges included limited internet connectivity and access barriers in remote field locations, particularly during major displacements and flooding. A lack of infrastructure and accountability mechanisms and reluctance by actors to share health information were also referenced, with one respondent from the private sector attributing this to a fear of repercussion by the government:

*"It's nothing other than this reluctance of sharing information…[Reluctance] across the board, data-sharing in general. There's no malice behind it. It's just there's not this thought behind the importance of transparency or visibility into what everyone is doing for fear of repercussions."* —Private Sector (KI11)

Together, these factors impacted the quality and comprehensiveness of data available and hindered the ability to access timely health information among both humanitarian and development actors, thus limiting effective coordination, decision-making, and prioritization of health needs. This phenomenon also translated to limitations in joint reporting between both actors and their ability to gain a cohesive and comprehensive picture of geographic presence by actor and respective health interventions implemented in each region. One donor interviewed elaborated on this point:

*"They're [donors and implementing agencies] not overlaying and speaking to each other, we need a really good information management unit to be able to take all these data sets and now start helping you to drill it down to make it much more accessible [but] that's real-time information that has to be updated quite regularly."*—Donor (KI21)

Numerous respondents indicated they were simply unaware of what other actors might be present in a geographic area if it fell outside of their mandate, be it humanitarian or development, primarily due to gaps in coordination among each other and delays in information-sharing. Although there is a Health Service Functionality Dashboard for South Sudan available which presents information (originally obtained from the District Health Information System 2), this was not updated in real-time [73].

**Field level coordination: a more feasible engagement amidst barriers at the national level.** In contrast to the limited engagement with the MoH at the national level, humanitarian and development actors at the sub-national level cited stronger coordination with the state and county MoH and the RRC. Though the RRC oversees humanitarian and disaster management at the sub-national level, both actors interviewed referenced coordination with this body and stressed that without coordination and clearance of planned interventions with both the state/county MoH and RRC, actors were unable to operate in a given geographic area. Despite compliance with such coordination protocols, there were still concerns around duplication in terms of health interventions, often leading to an excess or shortage of health services relative to population-based criteria for health services outlined by national health policies and global standards. One donor describes the saturation of services in Bentiu camp for internally displaced persons:

*"So, if I look at the example of [Bentiu] IDP camp, you have MSF, who's supporting the main hospital, Bentiu state hospital, supported by World Bank previously HPF. And then you have so many partners funded by all of us donors, [we're] completely guilty of it […] I think there are five PHCCs in [Bentiu] Camp, which is serving a population of 110,000 to 150,000, depending on the movement, but normally 110. […] But if you look at the national guidelines [for South Sudan and in accordance with global WHO criteria], it should be one PHCC for 50,000 people. So, then you're kind of going in and you're seeing duplication"*—Key Informant, Donor Agency (KI21)

Health committees in place within each community helped support coordination efforts for the PHCCs and PHCUs, however, the extent of their influence on different actors was less clear given the presence of different international actors who wielded more decision-making power. The need for both humanitarian and development actors to maintain their footprint in a given geographic area was mentioned, even at the risk of duplication or overlap.

## Discussion

This study demonstrates that there is not a clear distinction between humanitarian and development assistance in such a complex operational environment as South Sudan where there is an absence of peace. Consequently, there is a lack of clarity as to how the HDN can be operationalized, particularly in the more fragile states within South Sudan that face recurrent shocks and stresses. Key barriers identified through this research include how the nature of assistance by humanitarian and development actors was influenced by dramatic funding drawdowns for development health assistance, in addition to significant increases in humanitarian needs from 2022–2023, a lack of government capacity, and protracted internal conflict. Consequently, for many years, development assistance has been more humanitarian in nature, only implemented over a longer time period with funding from development donors; hence, our categorization of this as longer-term humanitarian assistance. Furthermore, there remain major development gaps in the health sector that have been exacerbated by recent funding drawdowns, which humanitarian funding has not, and should not, be expected to cover.

Recent funding drawdowns, particularly to the HPF, are part of an overall downward trend documented for development assistance to South Sudan over the past several years, and more broadly, on a global scale in 2023 [74,75]. These reductions prompted an examination of how the two largest development health funds in South Sudan (HPF and World Bank) may be administered more cohesively and equitably given the differences between the funds, as has been implemented in other contexts, such as Afghanistan's SEHAT project in 2013 and its successor the Sehatmandi project of 2018. Funded by the World Bank, SEHAT was also a shift from a split funding mechanism amongst multiple donors with divided coverage across different geographic regions of the country [76,77]. In a similar effort, a consolidation of the two development funds in South Sudan would also facilitate transition to the government in the long-term, in line with existing plans for development actors.

Following the period of our research, a consolidated multi-donor trust fund was in fact announced from 2024 onwards in place of the two separate health development projects. It will be called the new Health Sector Transformation Project, funded by the World Bank and delivered through UNICEF [78,79]. Although it is still too early to assess such a fund, its consolidation will likely enable more consistent and streamlined health services across the country. Efforts to enhance uniform approaches demonstrate the continued need for complementarity and coherence across and between humanitarian and development health assistance in South Sudan to use all available resources more efficiently. There should also be explicit and actionable transition plans between humanitarian and development interventions, particularly at the level of each state, that includes transparency in the funds provided and the interventions that will be implemented. Following the most recent and dire funding gaps, such concrete planning efforts are both timely and necessary. We would further recommend that these funding and intervention plans, together with how the transition from humanitarian to development will occur, be available on a website and accessible to the public to ensure transparency and accountability, particularly at the sub-national levels (both state and county) in South Sudan. Many of these recommendations align with innovative financing approaches that have been explored for South Sudan [75,80], and globally in other contexts (e.g., Ethiopia, Yemen) [81–83], such as multi-year financing for protracted crisis settings and cross-cutting assistance which integrates humanitarian funding within development assistance.

Engagement among the government as well as humanitarian and development actors in South Sudan has been extremely limited to date, particularly at the national level. This lack of coordination has contributed to limited decision-making power and leadership by the government. It is due to a number of reasons, including sanctions, lack of government transparency, and limited government contribution to the health sector budget despite promises to do so. While

targeted sanctions may address legitimate concerns around transparency and human rights violations in South Sudan, their indirect impact on the health sector has created an environment where the government has been inadequately involved in most aspects of the health planning and response, and in many instances, as our findings demonstrate, completely bypassed. A similar phenomenon was documented by Jones et al. (2015) and Sami et al. (2020) in South Sudan [20,75]. In those instances, the bulk of humanitarian and development assistance bypassed the government and was directly channeled to INGOs who provided the majority of health services in the country, though reasons for this were not attributed to sanctions, but rather due to a lack of capacity by the government (i.e., MoH) and according to Sami et al., "a lack of confidence by the international donor community in the peace process" [20,75]. Despite these restrictions, there have been some efforts underway to increase the engagement of the government at the state and county levels, though this has proven to be challenging for a variety of reasons including due to the centralization of the health system decision-making, despite national health policies and plans outlining a decentralized structure.

Beyond enhancing complementarity, humanitarian and development actors should also strengthen their engagement and decision-making processes with government counterparts, particularly health technocrats at the sub-national level (state, county, boma). While such enhanced engagement is important in all parts of the country, it may be particularly pertinent in specific geographic regions of the country that are deemed more stable than others, as such areas may be more conducive to sustainable development. Even with restrictions in place to engage at the national level in South Sudan, we believe these should not prevent coordination with local health authorities at the sub-national level. This level of sub-national engagement and service delivery has been feasible across multiple contexts, when there is a presence of sanctioned state and/or non-state actors (e.g., Syria, Yemen, Mali) [84–86]. This study also pointed to an overreliance on UN agencies and institutional donors (in addition to INGOs, as mentioned above) that subsidize and/or implement the majority of health services; over 62% of services in South Sudan were financed by humanitarian and development actors as of 2022 [22]. Such financial dependence has increased in the absence of sufficient investment in the national health budget by the government, and limited adherence to its financial commitments, despite the availability of government resources such as oil revenues [21,22]. The limited government contributions in South Sudan to the health sector are consistent with trends observed in other low-income countries where government prioritization of health spending has been shown to decrease when a significant amount of external aid is received. This suggests that external aid has replaced, rather than complemented, government spending according to the World Health Organization [87]. Ultimately, the discrepancies in implementation and funding by government and international actors in South Sudan may impact the viability of the health sector. Furthermore, this study suggests such reliance on external actors may continue to diminish the role of government in humanitarian health response and in its transition to longer-term, sustainable development.

A number of other factors influenced the feasibility of operationalizing the nexus, including rigid, structural silos between humanitarian and development agencies, especially multi-mandate agencies, contributing to separate streams of governance, financing, planning, programming, reporting, and coordination. The programming and financing aspects hold critical significance as humanitarian and development assistance are each governed by different life cycles, making it cumbersome to align interventions (both health and/or non-health) and find areas of complementarity. This alignment may be somewhat feasible within the same agency, as has been implemented in other contexts to date. This includes the European Union's interventions in Chad [88] and USAID programs in in Yemen [82,89] and the Sahel (Burkina Faso, Niger) [88]. But this form of integration between humanitarian and development assistance is still extremely challenging across different agencies, and at-scale, given the entrenched, bureaucratic structures that exist in such large institutions. Various bilateral and multilateral agencies have taken some strides to develop global (or regional) nexus policies, strategies, or guidance materials (though nexus terminology is not always explicitly used in these documents), including USAID [90,91], the European Union [92], and the United Kingdom Foreign, Commonwealth and Development Office [93].

However, there remains a need to develop, institutionalize, and disseminate practical and clear steps to adopt a nexus approach for health interventions in countries receiving significant amounts of humanitarian and development assistance,

including South Sudan. These headquarters' strategies should translate strategies into concrete standard operating procedures that guide how humanitarian and development actors can work together at regional and country levels, including clear and measurable process and outcome indicators. In South Sudan, this is essential for the largest international actors present who fund (donors) and/or deliver health services (UN, INGOs). Coordination platforms in South Sudan have also contributed to these silos (and to some redundancy in the numerous fora that exist) given the majority at the national and sub-national levels are either humanitarian or development. There were some exceptions where there were hybrid platforms (e.g., humanitarian-development or humanitarian-development-peace) such as the informal HDN coordination body. Despite the existence of this body and other ad hoc efforts, there were no joint mapping efforts by humanitarian and development actors in terms of charting activities implemented in each area of operation, thus limiting a clear and holistic picture of who was funding what and where for both forms of assistance. Although this was clear within the same agency, this mapping was separately conducted on the humanitarian side, and to a lesser degree on the development side for organizations operating in the same geographic region. Other high-level HDP platforms, such as the Partnership for Peace Recovery and Resilience were still in their infancy during the data collection period, so it is too premature to determine if such an initiative will garner enough political will and buy-in to propel nexus efforts forward. Additionally, some criticism has emerged that this specific initiative will further perpetuate a top-down approach, thus questioning if the localization agenda has been prioritized since LNGOs (and communities) have not led this effort relative to donors, UN agencies, and INGOs involved [94].

This research also found another coordination challenge regarding joint preparedness and planning between humanitarian and development actors (including the government) that negatively impacted health service delivery in the case of sudden emergencies. While both types of actors were able to activate contingency funds in these instances, such funds appeared to be primarily reactive rather than proactive or preventive in nature, despite the predictability of some shocks and stresses (e.g., seasonal, climate emergencies). Our review and stock-taking of rapid financing mechanisms demonstrated that there is some flexible funding available for both humanitarian and development actors to respond to emergencies, however, these remain separate, potentially limiting the efficiency of funds available and ability to flexibly transition between both forms of assistance. In the case of development actors (World Bank, HPF), cross-cutting emergency assistance suggested some integrated (nexus) efforts, however, such funds were often inadequate, as was the case with the HPF's annual contingency envelope (0.33%) relative to its entire funding portfolio. While it was assumed by some development actors that emergency funds could be activated via humanitarian actors, the recurrence and scale of these emergencies has made it more challenging to meet the increased needs in a timely, sufficient manner amidst a dwindling funding backdrop.

Lessons may be applied from existing efforts in South Sudan for more predictable emergencies like floods, which have entailed community preparedness measures implemented by the South Sudan Red Cross and preparedness funding released by the UN Central Emergency Response Fund and South Sudan Humanitarian Fund [95,96]. While these contingency measures were not health specific, they represented preparedness planning and resilience-building efforts for communities given the recurrence of climate shocks that inevitably impact health outcomes. Preparedness measures may also prevent further backsliding in health development gains, as had occurred during the COVID-19 pandemic, given the prioritization of emergency response (including over RMNCAH services in South Sudan and other low-income countries) may limit investment in health systems strengthening and resilience, both of which enable health systems and communities to absorb shocks and stresses and prevent backslides in health [20,97–101]. While there remain many challenges in strengthening health systems in South Sudan due to heightened and recurrent fragility, as was documented more recently by Jones et al., there remains a need to invest in such activities to help achieve longer-term development goals [20]. A systems approach should still be undertaken across all states, regardless of fragility, given the growing body of evidence on the need for health systems strengthening and resilience activities in fragile settings, as has been documented in South Sudan and other contexts to date, including Bangladesh, Yemen, Somalia, and the Democratic Republic of Congo

[102–104]. Finally, early recovery and longer-term development programming may be more feasible in the more stable states and counties relative to other fragile parts of the country. These areas may provide an opportunity to deliver more comprehensive services and pilot HDpN programming.

**Strengths and limitations.** This research had several strengths. The research fills a gap in the literature on the operationalization of health interventions in the nexus in South Sudan. This is the first study to examine the constructs of leadership and governance, financing, and coordination in depth as it pertains to the humanitarian-development-peace nexus and health interventions in South Sudan. The research included a wide range of perspectives from humanitarian and development practitioners in South Sudan, across multiple levels (e.g., donors, UN agencies, NGOs, health authorities, and the private sector). A rigorous methodology was used to develop the key informant interview guide, as the research team utilized the expertise of different subject matter experts (on both RMNCAH and South Sudan) to review and guide the process. Key findings were also reviewed by subject matter experts, including health experts from South Sudan and others with extensive knowledge of different components of the research to ensure a soundness and nuance of such findings (sanctions, coordination and governance structures, RMNCAH). The core research team consisted of humanitarian practitioners, with support from a team of health experts in South Sudan, in order to ensure a contextualized approach. The use of a case study to document findings of this research may prove useful for practitioners working in South Sudan and other fragile settings, as there have been limited examples documented in the literature to date.

The research also had some limitations. These include potential biases (e.g., selection, recall, and implicit), with efforts to limit these through ongoing reflexivity by researchers [52]. Given the research team was composed of experienced humanitarian practitioners, there may have been familiarity with some respondents interviewed. Since interviews often probed on sensitive topics, particularly those political in nature (e.g., governance, financing, sanctions), interviewees may have self-censored responses despite assurances of confidentiality. In addition, findings may have touched on topics beyond nexus and RMNCAH (or health) interventions, given the breadth of the HDN. While researchers focused on the specific topics in this paper, some respondents used examples beyond the research scope, often to illustrate priorities outside of, or in addition to, health, given the complexity and fragility in South Sudan. MoH officials in Juba were unable to meet the research team during the on-site visit and following, despite multiple attempts by the research team. This limited the ability to document firsthand perspectives among government stakeholders in Juba, forcing the researchers to rely on other insights from non-government personnel at the capital/national level in South Sudan. The research was commissioned by a USAID-funded RMNCAH program, so there may have been donor-specific considerations taken into account by the research team, including any current/past restrictions within RMNCAH interventions as outlined by US policies on foreign assistance for sexual and reproductive health. In addition to these limitations, access constraints restricted travel for the research team beyond Juba and Bor County for financial, logistical, and security reasons. While interviews conducted in Juba capital were intended to represent broader, country-level findings, this does not guarantee or signify country-wide phenomena. In addition, findings of this study are not nationally representative. Finally, this research did not include interviews with any direct recipients of health services (e.g., patients, clients), as researchers focused on stakeholders from the humanitarian and development sectors at this stage that were directly involved with the financing, design, and delivery of health interventions. A future area of focus may incorporate such perspectives given the importance of centering patients/clients in the design and delivery of services, and more broadly, in ensuring accountability to affected populations.

## Conclusion

Protracted fragility in South Sudan has made it challenging for humanitarian and development actors, including the government, to operationalize the nexus for health interventions. This study provides in-depth analysis and insight into the leadership and governance, financing, and coordination landscape in South Sudan, which impacts and influences the implementation of health services. The complex political and operational environment has contributed to significant

barriers to a coordinated, complementary, and coherent response by humanitarian and development health actors. Fractured engagement with the government has contributed to symbolic leadership and limitations in adequately involving MoH counterparts at the national level, and to a lesser extent, at sub-national levels. The shifting funding landscape for both development and humanitarian health actors provides an impetus for both actors to work together to use resources more efficiently. This will require greater investment and political will in existing platforms like the HDPN coordination body, Partnership for Peace, Recovery, and Resilience, and other more informal nexus discussions occurring in South Sudan. While the consolidation of funding has been one approach to streamline health service delivery among development actors, there is an imperative to integrate humanitarian and development funding and assistance beyond contingency funding mechanisms that currently exist. Humanitarian assistance should also incorporate more development-oriented activities, such as health systems strengthening and resilience efforts to ensure development gains are preserved and the health system and communities are able to absorb recurrent shocks and stresses, particularly given ongoing and heightened fragility in different parts of the country. Additionally, more sustainable interventions should be planned and funded, particularly in the areas of South Sudan that are deemed more stable and are primed for early recovery and more sustainable development activities. These areas may provide an opportunity to deliver more comprehensive services and pilot HDPN programming.

## Supporting information

**S1 File. S1 Table. Demographic Indicators in South Sudan, Regional (Sub-Saharan Africa), and Global Averages. S2 Table. Major Health Policies and Plans in South Sudan. S3 Fig. Timeline of Major Humanitarian and Development Funds in South Sudan. S4 Table. Documents Reviewed Prior to On-Site Data Collection.**
(ZIP)

## Acknowledgments

We thank all individuals interviewed in South Sudan for this research who generously provided their insights, knowledge, and expertise for the development of this case study. We thank the MOMENTUM and IMA World Health teams in South Sudan for their support throughout this research, including Mounir Lado, Lou Eluzai, Martha Awet, David Gai Deer, Aret Thony Aret, and Pitia Moses Lazarus; and Nancy Stroupe, Meghan Gallagher, and Melinda Pavin in Washington, DC. We thank Gathari Ndirangu Gichuhi and Abdelselam Daif for their technical review of the data collection tools for this research; Ahmad Odaimi and Okba Doghim for their technical guidance; and Yusra Shawar for her guidance during the data analysis phase. We thank Kemish Kenneth, Annette Hearns, Rachel Marcus, and Mark Ferullo for their review of research findings. Last, but certainly not least, we sincerely thank Shannon Doocy, Shannon Frattaroli, Hannah Tappis, and Andrea Wirtz from the Johns Hopkins Bloomberg School of Public Health for their guidance throughout this research and for their extensive review of the manuscript.

## Author contributions

**Conceptualization:** Amany Qaddour, Dan Wendo, Christopher Lindahl, Paul Spiegel.

**Data curation:** Amany Qaddour, Lauren Yan.

**Formal analysis:** Amany Qaddour, Lauren Yan.

**Funding acquisition:** Dan Wendo, Christopher Lindahl.

**Investigation:** Amany Qaddour, Lauren Yan, Paul Spiegel.

**Methodology:** Amany Qaddour, Paul Spiegel.

**Project administration:** Amany Qaddour, Lauren Yan, Laura Elisama, Paul Spiegel.

**Supervision:** Dan Wendo, Christopher Lindahl, Paul Spiegel.

**Validation:** Amany Qaddour, Lauren Yan, Paul Spiegel.

**Visualization:** Amany Qaddour.

**Writing – original draft:** Amany Qaddour.

**Writing – review & editing:** Amany Qaddour, Lauren Yan, Dan Wendo, Laura Elisama, Christopher Lindahl, Paul Spiegel.

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
