## [Decision Letter · Decision Letter 0]

28 Jan 2025

Dear Dr. Qaddour,

Thank you for submitting your manuscript to PLOS ONE. After careful consideration, we feel that it has merit but does not fully meet PLOS ONE’s publication criteria as it currently stands. Therefore, we invite you to submit a revised version of the manuscript that addresses the points raised during the review process.

We look forward to receiving your revised manuscript.

Kind regards,

Olushayo Oluseun Olu

Academic Editor

PLOS ONE

a) If there are ethical or legal restrictions on sharing a de-identified data set, please explain them in detail (e.g., data contain potentially identifying or sensitive patient information, data are owned by a third-party organization, etc.) and who has imposed them (e.g., a Research Ethics Committee or Institutional Review Board, etc.). Please also provide contact information for a data access committee, ethics committee, or other institutional body to which data requests may be sent

4. We note that Figure 2 in your submission contain [map/satellite] images which may be copyrighted. All PLOS content is published under the Creative Commons Attribution License (CC BY 4.0), which means that the manuscript, images, and Supporting Information files will be freely available online, and any third party is permitted to access, download, copy, distribute, and use these materials in any way, even commercially, with proper attribution. For these reasons, we cannot publish previously copyrighted maps or satellite images created using proprietary data, such as Google software (Google Maps, Street View, and Earth). For more information, see our copyright guidelines: http://journals.plos.org/plosone/s/licenses-and-copyright.

a. If you are unable to obtain permission from the original copyright holder to publish these figures under the CC BY 4.0 license or if the copyright holder’s requirements are incompatible with the CC BY 4.0 license, please either i) remove the figure or ii) supply a replacement figure that complies with the CC BY 4.0 license. Please check copyright information on all replacement figures and update the figure caption with source information. If applicable, please specify in the figure caption text when a figure is similar but not identical to the original image and is therefore for illustrative purposes only.

Reviewers' comments:

Reviewer's Responses to Questions

**Comments to the Author**

1. Is the manuscript technically sound, and do the data support the conclusions?

Reviewer #1: Partly

Reviewer #2: Yes

2. Has the statistical analysis been performed appropriately and rigorously?

Reviewer #1: No

Reviewer #2: Yes

3. Have the authors made all data underlying the findings in their manuscript fully available?

Reviewer #1: No

Reviewer #2: No

4. Is the manuscript presented in an intelligible fashion and written in standard English?

Reviewer #1: Yes

Reviewer #2: Yes

Reviewer #1: This study is relevant to the context of South Sudan because of the continuous lack of progress in achieving health targets despite huge investments. The paper is however more of a project report than a scientific paper that links findings to the literature. To improve the manuscript, the following are suggested:

1. Improve the introduction section with more literature review that defines the issues around the health-development-peace nexus. This will help provide a context to make more informed inferences from your findings in the discussions.

2. Be clearer what humanitarian and development means as related to health beyond the usual phrase "health system strengthening" which is often used without contextualization.

3. The key informants have different backgrounds with varying prejudices and biases. It will be useful to provide a clear analysis of the opinions of the different groups in the results section. This will provide a basis for a more objective analysis to find areas of agreement and divergence, which could be triangulated with the literature to formulate implementable recommendations.

4. There is too much focus on operationalizing the nexus focusing the three elements on Governance, financing and Coordination without being strongly supported by the how. The discussions should be stronger supported by evidence, including from the literature. There are many publications in the literature recommending how the nexus could be strengthened. How do they compare to the findings in South Sudan.

5.There are factors missing in the paper which could strengthen the arguments: a) How are the actors (Donors, Humanitarian, and Development) held accountable for all the resources being injected into the system vis a vis their performance. How could this be strengthened? b) Throughout the paper there is no linkage of the findings with the health indicators, except some mention of maternal and child mortality in the introduction session.

6. From 4 and 5 above, the discussions and conclusions should be strengthened by better articulating literature and findings, as well clearly articulating the benefits of doing and consequences of not doing so. This should be supported by the existing data from South Sudan on health indicators.

7. A limitation for this study which should be mentioned is non extension of the nexus to the peace element as Humanitarian-Development-Peace. A lot of the factors shaping the landscape in Juba are related to conflict and peace.

Reviewer #2: General: The article successfully demonstrated the impact of leadership and governance, financing, and coordination on the operationalization of health interventions in the humanitarian-development nexus in South Sudan. Although in Leadership and Governance: The limited government investment and engagement, compounded by sanctions and a lack of political will, have perpetuated reliance on international assistance and created barriers to effective coordination and sustainability of health services, article highlight the contributory economic and political shocks that serve as bedrock for this. Also in Financing, while significant reductions in development health funding have complicated progress towards long-term development objectives, leading to the closure of health facilities and increased reliance on humanitarian funding to fill gaps, article should demonstrate the crowding out effect of humanitarian funding to development funding with data. Under Coordination: Fragmented coordination among humanitarian and development actors, along with structural silos within multi-mandate agencies, has hindered efficient resource use and effective implementation of health interventions. Article should indicate reasons why there is limited political buy-in for the platform that is expected to serve better coordination.

Data: Author could consider anonymizing the key informants …KI1; KI2; KI3; KI4 ….Kin. Consider using number of KI whose opinion coalesced around major theme to strengthen discussion.

Specific comments:

• Authors to review lines 104 to 107: did those stated characteristics influence choice of study area; are they confounding factors?

• Authors should clarify line 141 vs 179…. interviews conducted onsite and use of zoom.

• Review line 498…the fact that there is no clear distinction between humanitarian and development assistance should be a facilitating factor to close the gap in HDN. Development workers should be able to draw synergy at planning to utilize funds to cover issues of development beyond humanitarian assistance.

**Do you want your identity to be public for this peer review?** For information about this choice, including consent withdrawal, please see our Privacy Policy

Reviewer #1: No

Reviewer #2: No

---

## [Author Response · Author response to Decision Letter 0]

6 Apr 2025

We would like to say thank you for the thoughtful reviews from the reviewers. They have been very helpful in improving our manuscript, and we are happy to submit a revised version.

In response to questions around data availability: The article analyzes qualitative, key informant interviews, we do not present numerical data. Due to the sensitive nature of the interviews, we cannot publish them online. Making interview notes publicly available would risk identifying participants, as the nature of responses includes context-specific details that could reveal identities. However, anonymized summarized qualitative interview data could be made available upon reasonable request, in line with ethical considerations and participant confidentiality. Please see the updated data availability statement which reflects this additional information. Please see the Response to Reviewers document along with the updated manuscript and another version of the manuscript with track changes.

---

## [Editor Report · Decision Letter 1]

17 Apr 2025

Leadership and Governance, Financing, and Coordination and their Impact on the Operationalization of Health Interventions in the Humanitarian-Development Nexus in South Sudan

PONE-D-24-45219R1

Dear Dr. Qaddour,

We’re pleased to inform you that your manuscript has been judged scientifically suitable for publication and will be formally accepted for publication once it meets all outstanding technical requirements.

Kind regards,

Olushayo Oluseun Olu

Academic Editor

PLOS ONE
---

## [Editor Report · Acceptance letter]

PONE-D-24-45219R1

PLOS ONE

Dear Dr. Qaddour,

I'm pleased to inform you that your manuscript has been deemed suitable for publication in PLOS ONE. Congratulations! Your manuscript is now being handed over to our production team.

Kind regards,

on behalf of

Dr. Olushayo Oluseun Olu

Academic Editor

PLOS ONE